# Risk stratification of atrial fibrillation and stroke using single nucleotide polymorphism and circulating biomarkers

**Tetsuo Sasano**[1]*, **Kensuke Ihara**[2], **Toshihiro Tanaka**[3,4], **Tetsushi Furukawa**[2]

**1** Department of Cardiovascular Medicine, Tokyo Medical and Dental University (TMDU), Tokyo, Japan, **2** Department of Bio-informational Pharmacology, Medical Research Institute, Tokyo Medical and Dental University (TMDU), Tokyo, Japan, **3** Bioresourse Research Center, Tokyo Medical and Dental University (TMDU), Tokyo, Japan, **4** Department of Human Genetics and Disease Diversity, Tokyo Medical and Dental University (TMDU), Tokyo, Japan

* sasano.bi@tmd.ac.jp, sasano.cvm@tmd.ac.jp

## Abstract

**Data Availability Statement:** All relevant data are within the paper and its Supporting Information files.

### Background

Atrial fibrillation (AF) is the most common sustained arrhythmia, and it causes a high rate of complications such as stroke. It is known that AF begins as paroxysmal form and gradually progresses to persistent form, and sometimes it is difficult to identify paroxysmal AF (PAF) before having stroke. The aim of this study is to evaluate the risk of PAF and stroke using genetic analysis and circulating biomarkers.

### Materials and methods

A total of 600 adult subjects were enrolled (300 from PAF and control groups). Peripheral blood was drawn to identify the genetic variation and biomarkers. Ten single nucleotide polymorphisms (SNPs) were analyzed, and circulating cell-free DNA (cfDNA) was measured from plasma. Four microRNAs (miR-99a-5p, miR-192-5p, miR-214-3p, and miR-342-5p) were quantified in serum using quantitative RT-PCR.

### Results

Genotyping identified 4 single nucleotide polymorphisms (SNPs) that were significantly associated with AF (rs6817105, rs3807989, rs10824026, and rs2106261), and the genetic risk score using 4 SNPs showed the area under the curve (AUC) of 0.631. Circulating miR-NAs and cfDNA did not show significant differences between PAF and control groups. The concentration of cfDNA was significantly higher in patients with a history of stroke, and the AUC was 0.950 to estimate the association with stroke.

### Conclusion

The risk of AF could be assessed by genetic risk score. Furthermore, the risk of stroke might be evaluated by plasma cfDNA level.

**Funding:** This study was performed by support from JRISTA (the support program for physician-initiated research by Bristol Myers Squibb). The funders had no role in study design, data collection and analysis, decision to publish, or preparation of the manuscript.

**Competing interests:** NO authors have competing interests.

## Background

Atrial fibrillation (AF) is the most frequent sustained arrhythmia, and it is responsible for high morbidity and mortality by several systemic complications [1]. Since the prevalence of AF increases with aging and other lifestyle diseases, it is urgent to establish a treatment strategy in an aging society. It is known that AF begins as paroxysmal form and gradually progresses to persistent form. Although the advancement in the catheter ablation strategy, AF with long-lasting history is refractory to treatment, and it might be beneficial to treat AF in its early phase [2].

Stroke is a leading cause of bedridden and disability and impairs the quality of life. Cardiogenic embolism is one of the most severe complications of AF and one of the most severe sub-types of stroke. Since the recovery of neurological function after severe stroke is still limited, it is important to predict the stroke and perform preventive therapy.

Several studies have been published to predict AF using various modalities, including genomic information, electrocardiogram, and biomarkers [3–8]. Recently, extracellular nucleotides have been attracting attention as novel biomarkers to be associated with AF [9–15].

Although these studies indicated the usefulness of genetic risk score and novel biomarkers to predict AF individually, it is still under investigation whether the combined analysis using genetic information and novel biomarkers increase the accuracy of prediction of AF. Thus, we aimed to pursue the estimation of AF and stroke by evaluation of single nucleotide polymorphism (SNP), and recently reported biomarkers including circulating miRNAs and cell-free DNA, and their combination.

Considering the final goal to establish the prediction of AF by blood test, the optimal study design was large-scale prospective study with promising items. To achieve the final goal, this study was positioned as the preliminary study using relatively small number of cases and cross-sectional design, to find promising items for predicting AF and stroke.

## Materials and methods

### Study subjects

A total of 600 adult subjects were enrolled in this study, which included healthy volunteers and patients who were referred to the Tokyo Medical and Dental University Hospital for the treatment of cardiovascular disease. The data collection began in Subjects were enrolled between April 2019 and August 2021. Exclusion criteria was as follows: age less than 50 years old, heart failure with continued drug treatment, rheumatic valvular disease, history of open-chest surgery, congenital heart disease, chronic inflammatory disease, malignant tumor, congenital coagulation factor dysfunction, hyperthyroidism, and hypothyroidism. The study was approved by the Ethics Committee of the Tokyo Medical and Dental University (No. G2000-188). All subjects were enrolled in this study with written informed consent at the timing of blood sampling. All measurements were performed in accordance with institutional regulations.

The enrolled subjects were divided into paroxysmal AF (PAF) group and control group. In this study, AF was defined as the electrocardiogram (ECG) recording over 30 seconds, and PAF was defined as AF that terminated spontaneously within 7 days. The control was defined as having no record of AF in ECG, no record of irregular pulse by physical examination in the annual health check, and no symptoms suggesting AF. We also extracted patients with a history of stroke from medical records, including all subtypes of cerebral infarction, but excluded cerebral and subarachnoid hemorrhage.

Since the aim of this study was to compare genetic information and novel biomarkers between PAF and control groups, we collected 300 subjects for each group.

## Blood sampling

The blood samples were collected from peripheral veins. The serum was obtained by centrifugation at 3000 rpm for 10 minutes, and stored at –80˚C until the RNA was isolated. The total RNA was extracted from the serum using mirVana™ PARIS™ Kit (Thermo Fisher Scientific, Santa Clara, CA) according to the manufacturer's instructions for the measurement of microRNA (miR). Synthetic Arabidopsis thaliana miR-159a (ath-miR-159a; 5′–UUUGGAUUGAAG GGAGCUCUA–3′) was spiked in to each sample for normalization. The expression of individual miRNAs (miR-99a-5p, miR-192-5p, miR-214-3p, and miR-342-5p) in serum was quantified by quantitative reverse transcriptase (RT)-PCR. Isolated total RNA samples (2 μL each) were reversely transcribed into cDNA and quantified using a TaqMan miR probe (Thermo Fisher Scientific). PCR reactions were performed on a Step One Realtime PCR system (Thermo Fisher Scientific). The Ct values for ath-miR-159a were used for normalization. The relative expression levels were calculated by the comparative Ct method ($2\Delta\Delta Ct$).

The blood sample for plasma and genomic analysis was collected using EDTA-2Na collection tube. The plasma was obtained by centrifugation at 3000 rpm for 10 minutes. Cell-free DNA (cfDNA) was extracted from 600 μL of plasma using MagMAX Cell-Free DNA Isolation Kit (Thermo Fisher Scientific, Carlsbad, CA, USA). The concentration of cfDNA was measured using Qubit dsDNA HS Assay Kit with Qubit 4.0 Fluorometer (Thermo Fisher Scientific). Genomic typing was performed in Macrogen Japan (Tokyo, Japan). In this study, we assessed 9 SNPs as shown in **Table 1**.

## Calculation of genetic risk score and CHADS$_2$ score

The genetic risk score (GRS) was calculated in two ways. GRS-1 was classified into 0 and 1 by selecting the dominant or recessive mode for each SNP and classifying them according to the presence or absence of risk. GRS-2 counted the number of risk alleles and was classified into 0, 1, and 2. Then, the scores for each SNP were summarized to obtain GRS.

The CHADS$_2$ score was calculated based on the clinical information as follows: congestive heart failure, hypertension, diabetes, and age > 74 years were counted for one point each, and history of stroke or transient ischemic attack (TIA) was counted as two points [16].

## Statistical analysis

Statistical analyses were performed with JMP®11 software (SAS Institute Inc., Cary, NC, USA). Data are expressed as mean ± standard deviation. Student's t-test was used for comparison between 2 continuous variables, unless described specifically. Receiver operating characteristic (ROC) analysis was performed to assess the accuracy of the parameters to discriminate

Table 1. Single nucleotide polymorphism for detection of AF.

| Target SNP | gene | major allele | minor allele |
|------------|--------|--------------|--------------|
| rs6666258 | KCNN3 | C | G |
| rs3903239 | PRRX1 | G | A |
| rs6817105 | PITX2 | C | T |
| rs3807989 | CAV1 | A | G |
| rs10821415 | C9orf3 | A | C |
| rs10824026 | SYNPO2L | G | A |
| rs1152591 | SYNE2 | A | G |
| rs7164883 | HCN4 | G | A |
| rs2106261 | ZFHX3 | T | C |

**Table 2. Characteristics of subjects.**

|  | PAF group | | | controls | | | p value |
|---|---|---|---|---|---|---|---|
| Age (yrs) | 70.3 | ± | 9.6 | 71.7 | ± | 9.9 | 0.08 |
| Male (n, %) | 179 | (59.7) | | 224 | (74.7) | | 0.01 |
| sBP (mmHg) | 130.4 | ± | 17.7 | 129.6 | ± | 17.6 | 0.58 |
| dBP (mmHg) | 75.2 | ± | 12.2 | 73.4 | ± | 12 | 0.07 |
| CHADS$_2$ score | 1.18 | ± | 0.94 | 1.52 | ± | 0.95 | 0.01 |
| Stroke (n, %) | 15 | (5.0) | | 16 | (5.3) | | 0.85 |
| NT-proBNP | 375 | ± | 1271 | 603 | ± | 2033 | 0.05 |
| CRP | 0.22 | ± | 0.1 | 0.22 | ± | 0.07 | 0.99 |

AF group, and the area under the curve (AUC) was calculated. A p <0.05 was considered statistically significant.

## Results

### Characteristics of subjects

Six hundred subjects were enrolled in this study, including 300 patients with PAF and 300 subjects with no history of AF. Characteristics of Patients are shown in **Table 2**. The average CHADS$_2$ score was 1.18±0.94 in PAF group and 1.52±0.95 in the control group. CHADS$_2$ score was higher in the control group.

### Single nucleotide polymorphisms related to AF

The results of genotyping are shown in **Table 3**. We found that 4 SNPs were significantly associated with AF (rs6817105, rs3807989, rs10824026, and rs2106261). Using these SNPs, we calculated the genetic risk score (GRS). GRS-1 was calculated with recessive mode for rs6817105 and rs3807989, and the dominant mode for rs10824026 and rs2106261, and summarized them, resulting in a score ranging 0 to 4. We also calculated GRS-2 by summarizing the number of risk alleles of the 4 SNPs, ranging 0 to 8. The ROC curve analyses are shown in **Fig 1**. GRS-1 showed the AUC of 0.626, sensitivity of 0.43, and specificity of 0.76. GRS-2 showed the AUC of 0.631, sensitivity of 0.51, and specificity of 0.68.

### Circulating biomarkers toin relation to the risk of AF

We then assessed the expression level of biomarkers including cell-free DNA (cfDNA) and 4 microRNAs. The concentration of cfDNA was 0.35±0.04 and 0.32±0.04 ng/μL in PAF and

**Table 3. Genotyping in relation to AF.**

| SNP | gene | PAF group | | | | | | control group | | | | | | p value |
|---|---|---|---|---|---|---|---|---|---|---|---|---|---|---|
|  |  |  | n |  | n |  | n |  | n |  | n |  | n |  |
| rs6666258 | KCNN3 | CC | 294 | GC | 6 | GG | 0 | CC | 291 | GC | 9 | GG | 0 | 0.4313 |
| rs3903239 | PRRX1 | AA | 93 | GA | 153 | GG | 54 | AA | 96 | GA | 149 | GG | 55 | 0.9466 |
| rs6817105 | PITX2 | CC | 68 | CT | 141 | TT | 91 | CC | 94 | CT | 150 | TT | 56 | **0.0016** |
| rs3807989 | CAV1 | AA | 152 | GA | 132 | GG | 16 | AA | 129 | GA | 142 | GG | 29 | **0.0483** |
| rs10821415 | C9orf3 | AA | 146 | CA | 128 | CC | 26 | AA | 138 | CA | 139 | CC | 23 | 0.6497 |
| rs10824026 | SYNPO2L | AA | 28 | AG | 163 | GG | 109 | AA | 50 | AG | 141 | GG | 109 | **0.0194** |
| rs1152591 | SYNE2 | AA | 121 | GA | 149 | GG | 30 | AA | 137 | GA | 125 | GG | 38 | 0.1326 |
| rs7164883 | HCN4 | AA | 3 | AG | 61 | GG | 236 | AA | 2 | AG | 62 | GG | 236 | 0.9006 |
| rs2106261 | ZFHX3 | CC | 46 | CT | 139 | TT | 115 | CC | 23 | CT | 114 | TT | 163 | **<0.0001** |

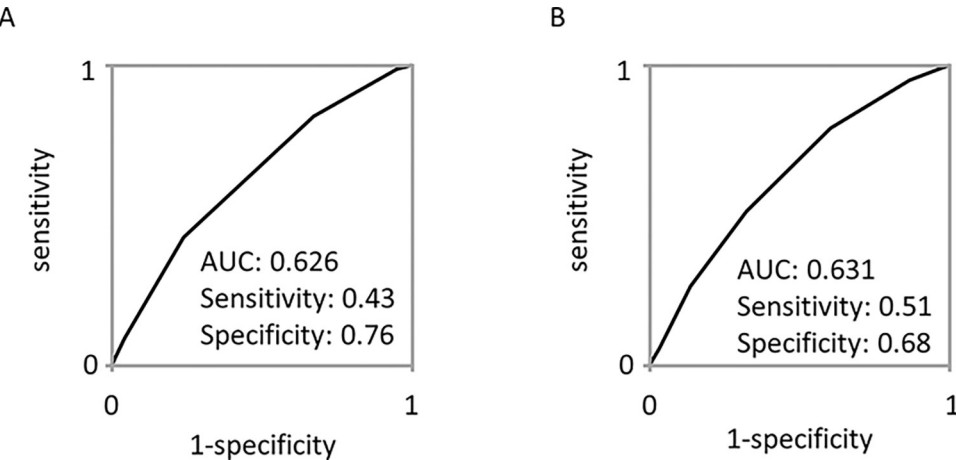

**Fig 1. Association between atrial fibrillation and genetic risk score.** ROC curves using GRS-1 (A) and GRS-2 (B) are shown.

control group, respectively (p = 0.53) (**Fig 2A**). Although PAF group tended to be lower expression in miR-99a-5p and higher expression in miR-342-5p, there was no significant difference between PAF and control groups (**Fig 2B**).

We also performed ROC curve analysis to estimate PAF group using 5 biomarkers. The area under the curve (AUC) was 0.562 in cfDNA, 0.538 in miR-99-5p, 0.529 in miR-192-5p, 0.531 in miR-214-3p, and 0.501 in miR-342-5p. All of these circulating biomarkers showed lower performance to estimate PAF group.

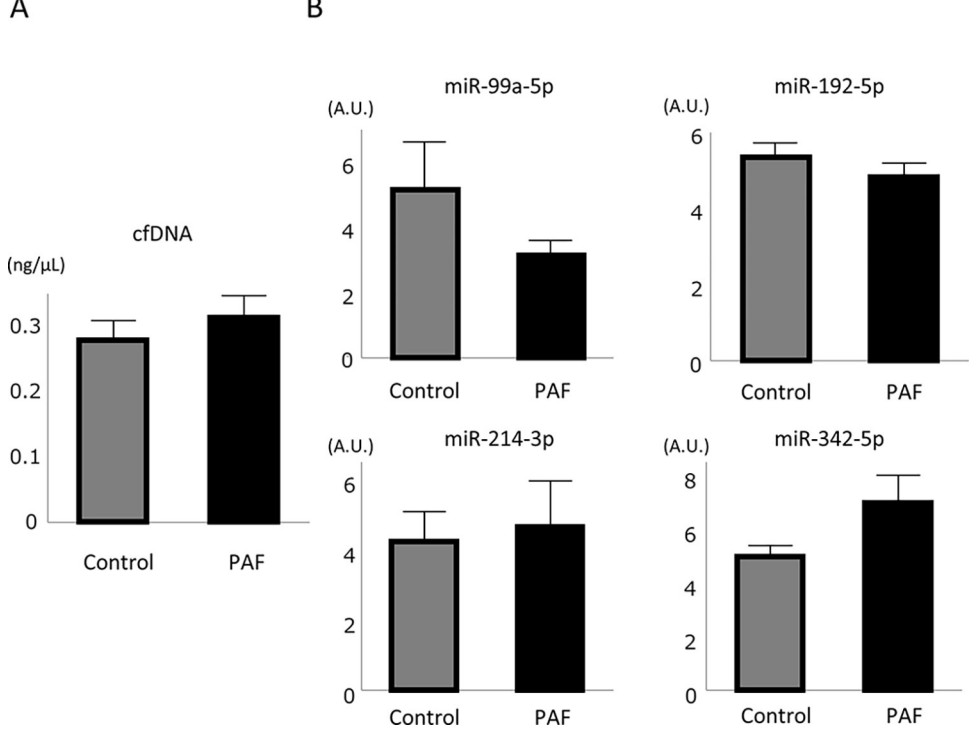

**Fig 2. Expression level of biomarkers.** (A) Cell-free DNA levels in plasma were higher in PAF group but did not reach to statistical significance. (B) Four previously reported miRNA was measured from the serum sample. All of them did not show statistical significance between PAF and control group.

**Table 4. Genotyping in relation to stroke.**

| SNP | gene | PAF group | | | | | | control group | | | | | | p value |
|---|---|---|---|---|---|---|---|---|---|---|---|---|---|---|
| | | | n | | n | | n | | n | | n | | n | |
| rs6666258 | KCNN3 | CC | 28 | GC | 3 | GG | 0 | CC | 557 | GC | 12 | GG | 0 | **0.0401** |
| rs3903239 | PRRX1 | AA | 10 | GA | 18 | GG | 3 | AA | 179 | GA | 284 | GG | 106 | 0.3814 |
| rs6817105 | PITX2 | CC | 9 | CT | 17 | TT | 5 | CC | 153 | CT | 274 | TT | 142 | 0.5049 |
| rs3807989 | CAV1 | AA | 14 | GA | 12 | GG | 5 | AA | 267 | GA | 262 | GG | 40 | 0.2390 |
| rs10821415 | C9orf3 | AA | 18 | CA | 9 | CC | 4 | AA | 266 | CA | 258 | CC | 45 | 0.1732 |
| rs10824026 | SYNPO2L | AA | 2 | AG | 14 | GG | 15 | AA | 76 | AG | 290 | GG | 203 | 0.2586 |
| rs1152591 | SYNE2 | AA | 16 | GA | 14 | GG | 1 | AA | 242 | GA | 260 | GG | 67 | 0.2077 |
| rs7164883 | HCN4 | AA | 1 | AG | 7 | GG | 23 | AA | 4 | AG | 116 | GG | 449 | 0.4729 |
| rs2106261 | ZFHX3 | CC | 6 | CT | 12 | TT | 13 | CC | 63 | CT | 241 | TT | 265 | 0.4264 |

## Risk stratification of stroke

In this study group, 31 patients had a history of stroke, 15 in PAF group and 16 in the control group. Genetic analysis using 10 SNPs are shown in **Table 4**. SNP in rs6666258 had a significant correlation with stroke (p = 0.04). We also analyzed circulating biomarkers, because these biomarkers might represent systemic inflammation and thrombogenesis. The plasma cfDNA level was significantly higher in subjects with stroke than without (2.14±2.03 vs. 0.24±0.40 ng/mL, p <0.01) (**Fig 3A**). When we analyzed the usefulness of cfDNA in PAF and control group separately, this difference was observed similarly (2.03±1.99 vs. 0.27±0.50 ng/mL, p <0.01, in PAF group, and 2.24±2.14 vs. 0.21±0.28 ng/mL, p <0.01, in control group). The ROC curve analysis showed AUC of 0.950, sensitivity of 1.0, and specificity of 0.79 (**Fig 3B**). We also evaluated the ROC curve analysis in PAF and control group separately, and found the AUC was similar (0.942 in PAF group, and 0.957 in control group). We also assessed microRNAs could estimate patients with stroke, which showed no significant difference in all 4 circulating microRNAs. Regarding the estimating performance of strokes, the AUC was 0.381 in miR-99-5p, 0.584 in miR-192-5p, 0.390 in miR-214-3p, and 0.615 in miR-342-5p.

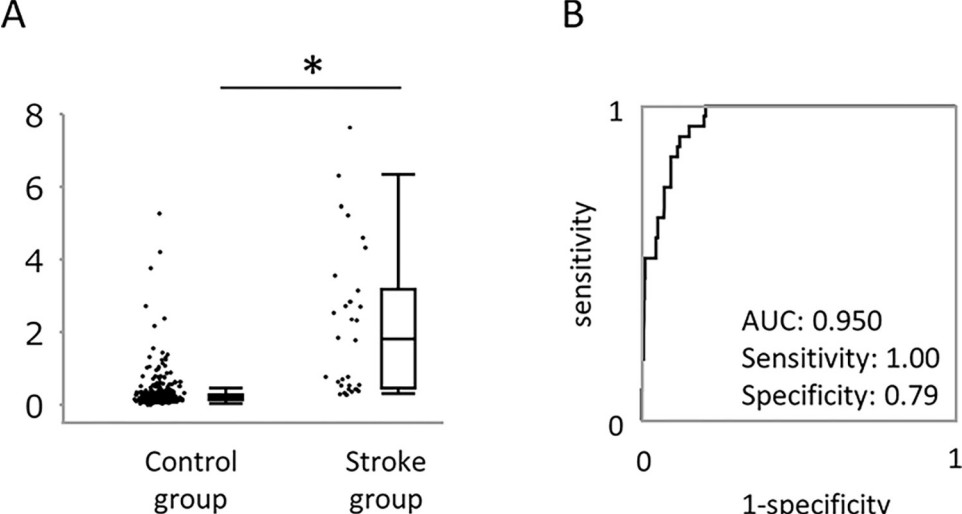

**Fig 3. Plasma cell-free DNA level as a biomarker to estimate subjects with stroke.** (A) The cfDNA levels were significantly larger in the stroke group than the control. (B) ROC curve analysis to estimate subjects with stroke. * p<0.01.

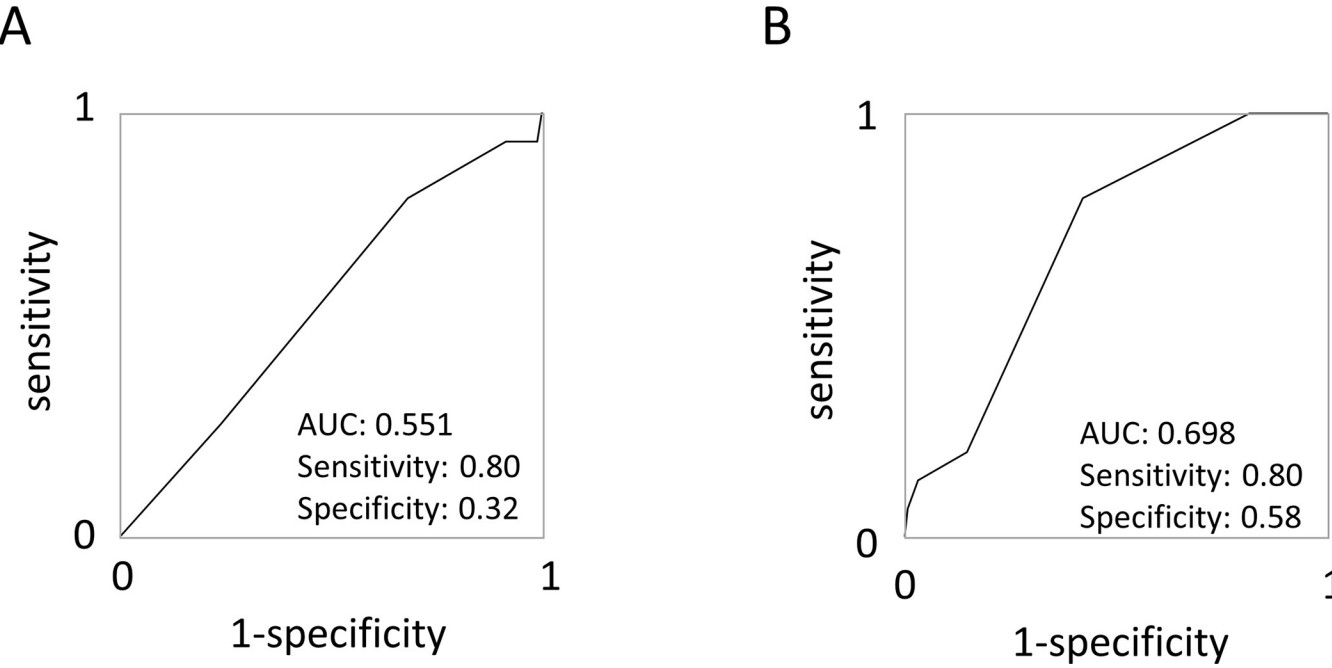

**Fig 4. Risk assessment for stroke using clinical information, genes, and biomarker.** (A) ROC curve analysis to estimate stroke group by CHADS$_2$ score in PAF group. (B) ROC curve analysis to estimate stroke group by combining CHADS$_2$ score, genetic risk score, and cfDNA score.

## Combined analysis of genetic risk and biomarkers in association with AF and stroke

We then combined the genetic risk and biomarkers to estimate AF or stroke. We generated a cfDNA risk score by using a cutoff value of 0.143 ng/mL. Combined AF or stroke risk scores were calculated with GRS-1 (ranging 0 to 4) or GRS-2 (ranging 0 to 8), and cfDNA risk score (ranging 0 or 1). The ROC curve analysis using this combined analysis showed AUC of 0.621 in GRS-1 with cfDNA score, and AUC of 0.628 in GRS-2 with cfDNA score.

Similarly, we combined the genetic risk score using rs6666258 (existence of C allele) and cfDNA score with a cutoff value of 0.270ng/mL, but this did not improve the estimation performance (AUC of 0.538).

For the risk stratification of stroke in AF patients, CHADS$_2$ or CHA$_2$DS$_2$-Vasc score has been widely utilized. We focused on the PAF group, and assessed whether the genetic risk and cfDNA improved the estimation accuracy in addition to CHADS$_2$ score. The ROC curve analysis using CHADS$_2$ score for estimation of stroke in PAF group showed the AUC of 0.551, sensitivity of 0.80, and specificity of 0.32 (**Fig 4A**). We generated genetic risk score for stroke (0 or 1) using the presence SNP (rs6666258), and cfDNA score (0 or 1) with optimal cutoff for cfDNA (0.28 ng/mL). Then we calculated combined risk score for stroke by summation of CHADS$_2$ score, genetic risk score, and cfDNA score. The ROC curve analysis showed the AUC of 0.698, sensitivity of 0.80, and specificity of 0.58 (**Fig 4B**).

## Discussion

In this study, we found 4 SNPs were related to AF in Japanese population. The nearest genes for these SNPs were PITX2, CAV1, ZFHX3, and SYNPO2L. Several GWAS studies identified high association between PITX2 and AF [3, 4, 17]. There are several explanations for the

mechanism by which PITX2 is involved in AF. PITX2 was known to be expressed in the left atrium but not in the right atrium, and it was essential for the formation of the pulmonary myocardial sleeve [18], which produced ectopic excitation to initiate AF [19]. In addition, PITX2 suppressed SHOX2, the transcription factor to generate pacemaker cells [20], which contributed to abolish pacemaker activity in the left atrium [21]. If the activity of PITX2 was increased, it facilitated the extension of pulmonary myocardial sleeve. On the other hand, if the activity of PITX2 was reduced, it might cause ectopic excitation in the left atrium. Both changes tended to initiate AF. ZFHX3 is known as having a strong association with AF in European and Asian populations [22, 23]. ZFHX3 interacts with the terminal end of protein inhibitor of activated STAT 3 (PIAS3), which is a specific inhibitor of signal transducer and activator of transcription 3 (STAT3). It was reported that L-type calcium channel had STATs response elements [24], and a positive feedback loop between STAT3 and microRNA might enhance the inflammation and fibrosis in the atrium [25]. Thus, the change in the activity of ZFHX3 may contribute to the electrical and structural remodeling in the atria. CAV1 played a role in maintaining the structure and function of caveolae [26]. CAV1 was reported to be associated with atrial fibrosis, and this process was partly via STAT3 signaling pathway [27]. The function of SYNPO2L in relation to the pathogenesis of AF is still under investigation. A recent study identified that the loss-of-function variant of SYNPO2L is associated with AF [28], probably with the impairment of the cytoskeleton. In this study, we tested 9 SNPs those were already known in association with AF, and we confirmed in 4 SNPs having significant association with AF in our relatively small population. These findings are considered as a basis for following analysis to combine genetic risk and biomarkers.Recently, several studies have been reported that circulating miR works as a biomarker to predict AF [9–13]. We also reported that miR diagnostic panel to predict AF [14]. However, the present study did not reach a statistical significance regarding candidate miRs (miR-99a-5p, miR-192-5p, miR-214-3p, and miR-342-5p). We also reported that mitochondrial cfDNA levels could predict PAF group [15]. Since the quantification of mitochondrial cfDNA required quantitative PCR and it might contain the technical error, we measured the total cfDNA level using a fluorometer in this study. The AUC of cfDNA to estimate PAF patients in this study was 0.562, which was lower than our previous study [15]. In this study, we collected blood samples without controlling the dietary condition of subjects. Recently, it was reported that the diurnal variation existed in the levels of microRNAs [29]. In addition, this study contained more patients with heart failure and a history of stroke in the control group than our previous studies. It might affect the level of circulating microRNAs and cfDNA in the controls.

Systemic complications including strokes were a critical issue in the treatment of AF. These various complications were occurred in relation to the systemic inflammation caused by AF. Although the precise mechanism that AF induced systemic inflammation has not been fully elucidated, we recently found that mitochondrial cell-free DNA released from the atrial myocyte induced an inflammatory response, and reported that circulating cfDNA worked as a biomarker to predict AF and the perpetuation of AF [15]. Considering the pathophysiology that cfDNA induces inflammation, it is assumed that cfDNA can be a biomarker for complications rather than AF itself. In this study, we compared the level of cfDNA in patients with a history of stroke or not, and we found a significant difference between them. Although several studies have been published to predict stroke [30–32], this study added cfDNA as a candidate for predicting marker of stroke.

The increase in cfDNA level was similarly found in PAF group and the control group, which might indicate that cfDNA was not a specific marker for cardiogenic embolism related with AF. Also, this is a cross-sectional study, and it is not clear whether increased cfDNA is the cause or result of strokes. This point needs to be clarified in future prospective studies.

In this study we evaluated the combined analysis for estimation of AF or stroke. Although the combined risk score did not show the improvement, the combined risk score using CHADS$_2$ score, genetic risk score, and cfDNA score increased the accuracy to estimate the association with stroke in PAF group.

## Limitations

This study has several limitations. First, the number of patients was relatively small. We did not find statistically significant differences in biomarkers, partly due to the small sample size. Second, we defined the control group having no record of AF in ECG, no record of irregular pulse by physical examination in the annual health check, and no symptoms suggesting AF. However, we cannot exclude the possibility that the control group contained undiagnosed PAF patients to some extent. It is difficult to completely exclude this possibility without life-long ECG monitoring. Third, this study was performed without a standardized protocol for the timing of blood sampling. This point affected the results of biomarkers as aforementioned. Third, this study is a cross-sectional study. Although we found that cfDNA level was higher in the stroke group, we have no clue whether the cfDNA was increased as a predictor or as a result of stroke. Prospective time-series studies will give us the answer.

## Conclusions

The prevalence of AF was associated with the genetic risk score. Although circulating biomarkers did not show the significant performance to improve the estimation of the AF group, the prevalence of stroke could be evaluated by plasma level of cfDNA.

## Supporting information

**S1 Checklist. *PLOS ONE* clinical studies checklist.**
(DOCX)

**S2 Checklist. STROBE statement—checklist of items that should be included in reports of observational studies.**
(DOCX)

**S1 Data.**
(XLSX)

## Author Contributions

**Conceptualization:** Tetsuo Sasano.

**Data curation:** Tetsuo Sasano, Kensuke Ihara.

**Supervision:** Toshihiro Tanaka, Tetsushi Furukawa.

**Writing – original draft:** Tetsuo Sasano.

**Writing – review & editing:** Tetsuo Sasano.

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
