## [Decision Letter · Decision Letter 0]

13 Jun 2023

PONE-D-23-13074Risk stratification of atrial fibrillation and stroke using single nucleotide polymorphism and circulating biomarkers

PLOS ONE

Dear Dr. Sasano,

Thank you for submitting your manuscript to PLOS ONE. After careful consideration, we feel that it has merit but does not fully meet PLOS ONE’s publication criteria as it currently stands. Therefore, we invite you to submit a revised version of the manuscript that addresses the points raised during the review process.

This study is designed to evaluate the risk of PAF and stroke using genetic analysis and circulating biomarkers. Although 600 subjects were enrolled and authors found that the concentration of cfDNA was significantly higher in patients with a history of stroke, the results of general impact is insufficient in current form. According to reviewer's comments, this paper should be revised.

We look forward to receiving your revised manuscript.

Kind regards,

Shinsuke Yuasa

Academic Editor

PLOS ONE

“This study was performed by support from JRISTA (the support program for physician-initiated research by Bristol Myers Squibb).”

Reviewers' comments:

Reviewer's Responses to Questions

**Comments to the Author**

1. Is the manuscript technically sound, and do the data support the conclusions?

Reviewer #1: No

Reviewer #2: Partly

2. Has the statistical analysis been performed appropriately and rigorously? 

Reviewer #1: No

Reviewer #2: I Don't Know

3. Have the authors made all data underlying the findings in their manuscript fully available?

Reviewer #1: No

Reviewer #2: Yes

4. Is the manuscript presented in an intelligible fashion and written in standard English?

Reviewer #1: Yes

Reviewer #2: Yes

5. Review Comments to the Author

Reviewer #1: This single-center study prospectively collected 600 blood samples from the refereed patients and healthy volunteers. The authors compared the presence of SNPs that have already been shown to contribute to the development of AF between participants with and without a history of paroxysmal AF and examined their diagnostic ability for both paroxysmal AF and Stroke in this population by ROC curves. The concept is interesting; however, the reviewer is concerned that the framework and methodology are not carefully described. Specific comments are as follows.

The most significant limitation of this study is that the authors collected blood samples from participants already diagnosed with AF or stroke and identified them. The relationship is unclear whether the morbidity caused the SNP mutation or vice versa. It is most desirable to predict future events from a disease-free state. Since several clinical risk scores (CHADS, CHA2DS-Vasc) are used in routine clinical practice to predict stroke, showing additive values of this genetic risk score is valuable.

This paper has two aims, to examine the validity of known SNPs associated with the development of AF and to investigate the relationship between these SNPs and a history of stroke. The Introduction needs to state why this study should be done clearly.

Various SNPs related to AF have already been reported. (Nat Genet. 2017 Jul 27;49(8):1286.) Still, the rationale for investigating these nine SNPs in this study must be clarified.

Definitions of critical variables should be described. It is unclear whether hemorrhagic infarction is included, the definition of paroxysmal, and the basis for the diagnosis.

How were these 600 participants recruited? If the authors were initially designed to compare 300 participants, each paroxysmal AF vs. non-AF, that should be stated.

Reviewer #2: The authors aimed to evaluate the risk of paroxysmal AF (PAF) and stroke using genetic analysis and circulating biomarkers. They enrolled 600 adult subjects (300 PAF and 300 controls), and identified the genetic variation and biomarkers. The results showed genotyping identified 4 SNPs being associated with AF (rs6817105, rs3807989, rs10824026, and rs2106261). Circulating miRNAs and cfDNA did not show significant differences between PAF and control groups, but the concentration of cfDNA was higher in patients with a history of stroke. The authors concluded that the risk of PAF could be assessed by genetic risk score, and the risk of stroke may be evaluated by plasma cfDNA level. Although the data are interesting, there are several concerns.

First, the study consisted of a small number of PAF and control subjects. As indicated by the authors, early detection of PAF seems to be useful for early intervention that prevents progression of PAF to persistent AF. From the present small sample study, only a preliminary data may be obtained. How do the authors confirm that control group did not include PAF? Considering the similar clinical characteristics between the 2 groups, it may be difficult to exclude the possibility that control group includes some asymptomatic PAF patients.

Second, 4 SNPs they identified are already said to be associated with AF. What are the new findings shown in this study? Please clarify this.

Third, they found that the concentration of cfDNA was higher in patients with a history of stroke. Was this specific for PAF group? Which type of stroke was associated with the increased cfDNA level? Was it a cardioembolic stroke? Was the increase a cause or result? Please clarify this.

Finally, this reviewer understands that this is a preliminary study for the large-size, prospective study in which the relation of genetic variation found in the study will be tested in a large number of subjects. If so, please state clearly in the manuscript.

6. PLOS authors have the option to publish the peer review history of their article (what does this mean?). If published, this will include your full peer review and any attached files.

Reviewer #1: No

Reviewer #2: No

---

## [Author Response · Author response to Decision Letter 0]

4 Aug 2023

Reviewer #1: This single-center study prospectively collected 600 blood samples from the refereed patients and healthy volunteers. The authors compared the presence of SNPs that have already been shown to contribute to the development of AF between participants with and without a history of paroxysmal AF and examined their diagnostic ability for both paroxysmal AF and stroke in this population by ROC curves. The concept is interesting; however, the reviewer is concerned that the framework and methodology are not carefully described. Specific comments are as follows.

The most significant limitation of this study is that the authors collected blood samples from participants already diagnosed with AF or stroke and identified them. The relationship is unclear whether the morbidity caused the SNP mutation or vice versa. It is most desirable to predict future events from a disease-free state. Since several clinical risk scores (CHADS, CHA2DS-Vasc) are used in routine clinical practice to predict stroke, showing additive values of this genetic risk score is valuable.

We appreciate the reviewers’ valuable comments. 

As the reviewer pointed out, the design of this study is a cross-sectional study comparing patients with AF with controls. Regarding SNPs, the genetic polymorphisms are determined at birth and do not change, thus we can exclude the possibility that AF cause SNP mutations. In terms of miRNAs and cell-free DNA, it is not clear to say the increased level of biomarkers is the cause or result of AF or strokes. We totally agree with the reviewer that a large-scale prospective study is required to answer this issue. This study is positioned as the preliminary study for coming large-scale prospective study. We added this point in the revised manuscript. 

In addition, according to the reviewer’s suggestion, we calculated the risk score for stroke in PAF group, by combining CHADS2 score, genetic risk score, and cfDNA score. We added these results as Fig 4 in revised manuscript.

This paper has two aims, to examine the validity of known SNPs associated with the development of AF and to investigate the relationship between these SNPs and a history of stroke. The Introduction needs to state why this study should be done clearly.

Thank you for the comments regarding critical point. Our working hypothesis is the combined analysis including genetic information and circulating biomarkers contributed to increase the accuracy of the prediction of AF and stroke. And this cross-sectional study might work to choose promising items for prediction of AF or stroke. We added the sentence for the aim of this study.

Various SNPs related to AF have already been reported. (Nat Genet. 2017 Jul 27;49(8):1286.) Still, the rationale for investigating these nine SNPs in this study must be clarified.

As aforementioned, the main purpose of this study is to investigate if the integrated analysis combining SNPs and biomarkers improve the prediction accuracy for AF and strokes. Thus we chose SNPs which were already known in association with AF in Japanese population. It is out of our scope to find novel SNP in relation to the AF. We stated this point in revised manuscript.

Definitions of critical variables should be described. It is unclear whether hemorrhagic infarction is included, the definition of paroxysmal, and the basis for the diagnosis.

Thank you for raising important points. We included hemorrhagic infarction in stroke group but excluded cerebral and subarachnoid hemorrhage. We added that with the definition of strokes and paroxysmal AF, and the diagnostic basis for them.

How were these 600 participants recruited? If the authors were initially designed to compare 300 participants, each paroxysmal AF vs. non-AF, that should be stated.

As we stated previously, this study is positioned as preliminary study to pick up the measurement items for coming large-scale prospective analysis. Thus we picked up each 300 participants for PAF and control groups. We added this description.

 

Reviewer #2: The authors aimed to evaluate the risk of paroxysmal AF (PAF) and stroke using genetic analysis and circulating biomarkers. They enrolled 600 adult subjects (300 PAF and 300 controls), and identified the genetic variation and biomarkers. The results showed genotyping identified 4 SNPs being associated with AF (rs6817105, rs3807989, rs10824026, and rs2106261). Circulating miRNAs and cfDNA did not show significant differences between PAF and control groups, but the concentration of cfDNA was higher in patients with a history of stroke. The authors concluded that the risk of PAF could be assessed by genetic risk score, and the risk of stroke may be evaluated by plasma cfDNA level. Although the data are interesting, there are several concerns.

First, the study consisted of a small number of PAF and control subjects. As indicated by the authors, early detection of PAF seems to be useful for early intervention that prevents progression of PAF to persistent AF. From the present small sample study, only a preliminary data may be obtained. How do the authors confirm that control group did not include PAF? Considering the similar clinical characteristics between the 2 groups, it may be difficult to exclude the possibility that control group includes some asymptomatic PAF patients.

Thank you for valuable comments. As the reviewer pointed out, the large limitation of this study is its relatively small sample size. We described this point in limitation.

 We defined the control group as having no record of AF in ECG, no record of irregular pulse by physical examination in the annual health check, and no symptoms suggesting AF. However, as the reviewer indicated, we cannot exclude the possibility that the control group might have asymptomatic AF episodes. We added the definition of control group in the Methods, and the description of limitation. 

Second, 4 SNPs they identified are already said to be associated with AF. What are the new findings shown in this study? Please clarify this.

Thank you for the comments regarding critical point. As the reviewer indicated, all 4 SNPs are already known to be associated with AF in previous studies, and we have no new findings focusing only for SNPs. Our working hypothesis is the combined analysis including genetic information and circulating biomarkers contributed to increase the accuracy of the prediction of AF or stroke. And this cross-sectional study works to choose possible items for prediction of AF. We added the sentence for the aim of this study.

Third, they found that the concentration of cfDNA was higher in patients with a history of stroke. Was this specific for PAF group? Which type of stroke was associated with the increased cfDNA level? Was it a cardioembolic stroke? Was the increase a cause or result? Please clarify this.

Thank you for raising up the critical point. The level of cfDNA was similarly higher in the stroke group both in the PAF group and control group. And we extracted the information of the history of stroke from medical record. Thus these results indicated that cfDNA was the biomarker for stroke including all subtypes, but not a specific biomarker for cardiogenic stroke. 

And we appreciate to give us another important point. Since this is the cross-sectional study, we cannot say the increased level of cfDNA is a cause or result. We need to clarify this with prospective study. We added the data this in the discussion in revised manuscript.

Finally, this reviewer understands that this is a preliminary study for the large-size, prospective study in which the relation of genetic variation found in the study will be tested in a large number of subjects. If so, please state clearly in the manuscript.

We really appreciate the reviewer’s important suggestion. The understanding of reviewer is totally correct. This paper is the preliminary study to choose items for coming large-scale study. We added the description in the revised manuscript.

---

## [Decision Letter · Decision Letter 1]

16 Aug 2023

PONE-D-23-13074R1Risk stratification of atrial fibrillation and stroke using single nucleotide polymorphism and circulating biomarkersPLOS ONE

Dear Dr. Sasano,

Thank you for submitting your manuscript to PLOS ONE. After careful consideration, we feel that it has merit but does not fully meet PLOS ONE’s publication criteria as it currently stands. Therefore, we invite you to submit a revised version of the manuscript that addresses the points raised during the review process.

We look forward to receiving your revised manuscript.

Kind regards,

Shinsuke Yuasa

Academic Editor

PLOS ONE

Journal Requirements:

Reviewers' comments:

Reviewer's Responses to Questions

**Comments to the Author**

1. If the authors have adequately addressed your comments raised in a previous round of review and you feel that this manuscript is now acceptable for publication, you may indicate that here to bypass the “Comments to the Author” section, enter your conflict of interest statement in the “Confidential to Editor” section, and submit your "Accept" recommendation.

Reviewer #1: All comments have been addressed

Reviewer #2: All comments have been addressed

2. Is the manuscript technically sound, and do the data support the conclusions?

Reviewer #1: No

Reviewer #2: Yes

3. Has the statistical analysis been performed appropriately and rigorously? 

Reviewer #1: No

Reviewer #2: Yes

4. Have the authors made all data underlying the findings in their manuscript fully available?

Reviewer #1: Yes

Reviewer #2: Yes

5. Is the manuscript presented in an intelligible fashion and written in standard English?

Reviewer #1: Yes

Reviewer #2: Yes

6. Review Comments to the Author

Reviewer #1: The authors correctly addressed the comments, and the manuscript has been improved. However, the reviewer still wants to mention a few points.

The authors often used the phrase "prediction of AF or stroke," but this is misleading and only looks at the association with AF or stroke, given the nature of a retrospective cross-sectional study. Additionally, from the results of this paper, the authors should not conclude that "the risk of AF could be assessed by combining genetic risk scores." The authors should correct this throughout the paper.

Detailed information, the calculation for genetic risk score, and the combined risk score for stroke by CHADS2 and genetic risk score (some readers will not understand what CHADS2 stands for) should be included in the Method section rather than the Results.

Reviewer #2: Thank you for revising the manuscript according to the comments of this reviewer. Although there are limitations in this study, it would be useful for further investigation in a large number of patients. I do not have any other comments to this manuscript.

7. PLOS authors have the option to publish the peer review history of their article (what does this mean?). If published, this will include your full peer review and any attached files.

Reviewer #1: No

Reviewer #2: No

---

## [Author Response · Author response to Decision Letter 1]

11 Sep 2023

We appreciate the reviewers for taking time to review our manuscript and giving valuable comments.

Reviewer #1: The authors correctly addressed the comments, and the manuscript has been improved. However, the reviewer still wants to mention a few points.

The authors often used the phrase "prediction of AF or stroke," but this is misleading and only looks at the association with AF or stroke, given the nature of a retrospective cross-sectional study. Additionally, from the results of this paper, the authors should not conclude that "the risk of AF could be assessed by combining genetic risk scores." The authors should correct this throughout the paper.

Thank you for raising the point. We understand that the phrase "prediction of AF or stroke" is inadequate for this retrospective cross-sectional study. According to the reviewer’s suggestion, we changed that phrase to “estimate the association with AF or stroke” or “estimate　the AF or stroke group” throughout the manuscript. We also removed the phrase "the risk of AF could be assessed by combining genetic risk scores" in conclusion, and changed the entire paragraph. We remained using “predict/prediction” in the context of our final goal and the description of reference.

Detailed information, the calculation for genetic risk score, and the combined risk score for stroke by CHADS2 and genetic risk score (some readers will not understand what CHADS2 stands for) should be included in the Method section rather than the Results.

Thank you for the valuable comment. According to the reviewer’s comment, we added the definition and calculation formula of CHADS2 score and genetic risk score in Method section. We added the reference to explain CHADS2 score in revised manuscript.

Reviewer #2: Thank you for revising the manuscript according to the comments of this reviewer. Although there are limitations in this study, it would be useful for further investigation in a large number of patients. I do not have any other comments to this manuscript.

We really appreciate the reviewer’s valuable suggestions and comments. Thanks to the reviewer, we consider that our manuscript improved for taking time to review our manuscript and giving valuable comments.

---

## [Decision Letter · Decision Letter 2]

13 Sep 2023

Risk stratification of atrial fibrillation and stroke using single nucleotide polymorphism and circulating biomarkers

PONE-D-23-13074R2

Dear Dr. Sasano,

We’re pleased to inform you that your manuscript has been judged scientifically suitable for publication and will be formally accepted for publication once it meets all outstanding technical requirements.

Kind regards,

Shinsuke Yuasa

Academic Editor

PLOS ONE

Additional Editor Comments (optional):

Reviewers' comments:

Reviewer's Responses to Questions

**Comments to the Author**

1. If the authors have adequately addressed your comments raised in a previous round of review and you feel that this manuscript is now acceptable for publication, you may indicate that here to bypass the “Comments to the Author” section, enter your conflict of interest statement in the “Confidential to Editor” section, and submit your "Accept" recommendation.

Reviewer #1: All comments have been addressed

2. Is the manuscript technically sound, and do the data support the conclusions?

Reviewer #1: Yes

3. Has the statistical analysis been performed appropriately and rigorously? 

Reviewer #1: Yes

4. Have the authors made all data underlying the findings in their manuscript fully available?

Reviewer #1: Yes

5. Is the manuscript presented in an intelligible fashion and written in standard English?

Reviewer #1: Yes

6. Review Comments to the Author

Reviewer #1: The reviewer appreciates the extensive responses to the comments and believes the manuscript has been improved.

7. PLOS authors have the option to publish the peer review history of their article (what does this mean?). If published, this will include your full peer review and any attached files.

Reviewer #1: No

---

## [Editor Report · Acceptance letter]

4 Oct 2023

PONE-D-23-13074R2 

Risk stratification of atrial fibrillation and stroke using single nucleotide polymorphism and circulating biomarkers 

Dear Dr. Sasano:

I'm pleased to inform you that your manuscript has been deemed suitable for publication in PLOS ONE. Congratulations! Your manuscript is now with our production department. 

Kind regards, 

on behalf of

Dr. Shinsuke Yuasa 

Academic Editor

PLOS ONE